# Transductive Non-linear Learning for Chinese Hypernym Prediction

## Abstract

Finding the correct hypernyms for entities is essential for taxonomy learning, fine-grained entity categorization, knowledge base construction, etc. Due to the flexibility of the Chinese language, it is challenging to identify hypernyms in Chinese accurately. Rather than extracting hypernyms from texts, in this paper, we present a transductive learning approach to establish mappings from entities to hypernyms in the embedding space directly. It combines linear and non-linear embedding projection models, with the capacity of encoding arbitrary language-specific rules. Experiments on real-world datasets illustrate that our approach outperforms previous methods for Chinese hypernym prediction.

## 1 Introduction

A hypernym of an entity characterizes the *type* or the *class* of the entity. For example, the word `country` is the hypernym of the entity `Canada`. The accurate prediction of hypernyms benefits a variety of NLP tasks, such as taxonomy learning (Wu et al., 2012; Fu et al., 2014), fine-grained entity categorization (Ren et al., 2016), knowledge base construction (Suchanek et al., 2007), etc.

In previous work, the detection of hypernyms requires lexical, syntactic and/or semantic analysis of relations between entities and their respective hypernyms from a language-specific knowledge source. For example, Hearst (1992) is the pioneer work to extract *is-a* relations from a text corpus based on handcraft patterns. The following-up work mostly focuses on *is-a* relation extraction using automatically generated patterns (Snow et al., 2004; Ritter et al., 2009; Sang and Hof-

mann, 2009; Kozareva and Hovy, 2010) and relation inference based on distributional similarity measures (Kotlerman et al., 2010; Lenci and Benotto, 2012; Shwartz et al., 2016).

While these approaches have relatively high precision over English corpora, extracting hypernyms for entities is still challenging for Chinese. From the linguistic perspective, Chinese is a lower-resourced language with very flexible expressions and grammatical rules (Wang et al., 2015). For instance, there are no word spaces, explicit tenses and voices, and distinctions between singular and plural forms in Chinese. The order of words can be changed flexibly in sentences. Hence, as previous research indicates, hypernym extraction methods for English are not necessarily suitable for the Chinese language (Fu et al., 2014; Wang et al., 2015; Wang and He, 2016).

Based on such conditions, several classification methods are proposed to distinguish *is-a* and *not-is-a* relations based on Chinese encyclopedias (Lu et al., 2015; Li et al., 2015). Similar to Princeton WordNet, a few Chinese wordnets have also been developed (Huang et al., 2004; Xu et al., 2008; Wang and Bond, 2013). The most recent approaches for Chinese *is-a* relation extraction (Fu et al., 2014; Wang and He, 2016) use word embedding based linear projection models to map embeddings of hyponyms to those of their hypernyms, which outperform previous algorithms.

However, we argue that these projection-based methods may have three potential limitations: (i) Only positive *is-a* relations are used for projection learning. The distinctions between *is-a* and *not-is-a* relations in the embedding space are not modeled. (ii) These methods lack the capacity to encode linguistic rules, which are designed by linguists and usually have high precision. (iii) It assumes that the linguistic regularities of *is-a* relations can be solely captured by single or multiple

linear projection models.

In this paper, we address these limitations by a two-stage transductive learning approach. It distinguishes *is-a* and *not-is-a* relations given a Chinese word/phrase pair as input. In the initial stage, we train linear projection models on positive and negative training data separately and predict *is-a* relations jointly. In the transductive learning stage, the initial prediction results, linguistic rules and the non-linear mappings from entities to hypernyms are optimized simultaneously in a unified framework. This optimization problem can be efficiently solved by blockwise gradient descent. We evaluate our method over two public datasets and show that it outperforms state-of-the-art approaches for Chinese hypernym prediction.[1]

The rest of this paper is organized as follows. We summarize the related work in Section 2. Our approach is introduced in Section 3. Experimental results are presented in Section 4. We conclude our paper in Section 5.[2]

## 2 Related Work

In this section, we overview the related work on hypernym prediction and discuss the challenges of Chinese hypernym detection.

**Pattern based methods** identify *is-a* relations from texts by handcraft or automatically generated patterns. Hearst patterns (Hearst, 1992) are lexical patterns in English that are employed to extract *is-a* relations for taxonomy construction (Wu et al., 2012). Automatic approaches mostly use iterative learning paradigms such that the system learns new *is-a* relations and patterns simultaneously. A few relevant studies can be found in (Caraballo, 1999; Etzioni et al., 2004; Sang, 2007; Pantel and Pennacchiotti, 2006; Kozareva and Hovy, 2010). To avoid "semantic drift" in iterations, Snow et al. (2004) train a hypernym classifier based on syntactic features based on parse trees. Carlson et al. (2010) exploit multiple learners to extract relations via coupled learning. These approaches are

---

[1] While there is abundant research on hypernym prediction for English with high precision, we mostly focus on the Chinese language in this paper. However, the proposed method is not entirely language-specific and has the potential to be adapted to other languages. In Section 4, we provide additional experiments to show that our approach also outperforms several existing methods for hypernym prediction in the English environment. We also discuss some potential applications of our method.

[2] Additionally, we present more details on algorithms and experimental settings and a prototype system for taxonomy visualization in the supplementary notes.

not effective for Chinese for two reasons: i) Chinese *is-a* relations are expressed in a highly flexible manner (Fu et al., 2014) and ii) the accuracy of basic NLP tasks such as dependency parsing still need improvement for Chinese (Li et al., 2013).

**Inference based methods** take advantage of distributional similarity measures (DSM) to infer relations between words. They assume that a hypernym may appear in all contexts of the hyponyms and a hyponym can only appear in part of the contexts of its hypernyms. In previous work, Kotlerman et al. (2010) design directional DSMs to model the asymmetric property of *is-a* relations. Other DSMs are introduced in (Bhagat et al., 2007; Szpektor et al., 2007; Lenci and Benotto, 2012; Santus et al., 2014). Shwartz et al. (2016) combine dependency parsing and DSM to improve the performance of hypernymy detection. The reason why DSM is not effective for Chinese is that the contexts of entities in Chinese are flexible and sparse.

**Encyclopedia based methods** take encyclopedias as knowledge sources to construct taxonomies. Ponzetto and Strube (2007) design features from multiple aspects to predict *is-a* relations between entities and categories in English Wikipedia. The taxonomy in YAGO (Suchanek et al., 2007) is constructed by linking conceptual categories in Wikipedia to WordNet synsets (Miller, 1995). For Chinese, Li et al. (2015) propose an SVM-based approach to build a large Chinese taxonomy from Wikipedia. Similar classification based algorithms are presented in (Fu et al., 2013; Lu et al., 2015). Due to the lack of Chinese version of WordNet, several Chinese semantic dictionaries have been conducted, such as Sinica BOW (Huang et al., 2004), SEW (Xu et al., 2008), COW (Wang and Bond, 2013), etc. These approaches have higher accuracy than mining hypernym relations from texts directly. However, they heavily rely on existing knowledge sources and are difficult to extend to different domains.

To tackle these challenges, **word embedding based methods** directly model the task of hypernym prediction as learning a mapping from entity vectors to their respective hypernym vectors in the embedding space. The vectors can be pre-trained by neural language models (Mikolov et al., 2013). For the Chinese language, Fu et al. (2014) train piecewise linear projection models based on a Chinese thesaurus. The state-of-the-art method

(Wang and He, 2016) combines an iterative learning procedure and Chinese Hearst-style patterns to improve the performance of projection models. They can reduce data noise by avoiding direct parsing of Chinese texts, but still capture the linguistic regularities of *is-a* relations based on word embeddings. Additionally, several work aims to study how to combine word embeddings for relation classification, such as (Mirza and Tonelli, 2016). In our paper, we extend these approaches by modeling non-linear mappings from entities to hypernyms and adding linguistic rules via a unified transductive learning framework.

## 3 Proposed Approach

This section begins with a brief overview of our approach. After that, the detailed steps and the learning algorithm are introduced in detail.

### 3.1 Overview

Given a word/phase pair $(x_i, y_i)$, the goal of our task is to learn a classification model to predict whether $y_i$ is the hypernym of $x_i$.

As illustrated in Figure 1, our approach has two stages: initial stage and transductive learning stage. The input is a positive *is-a* set $D^+$, a negative *is-a* set $D^-$ and an unlabeled set $D^U$, all of which are the collections of word/phrase pairs.

Denote $\mathbf{x}_i$ as the embedding vector of word $x_i$, pre-trained and stored in a lookup table. In the initial stage, we train a linear projection model over $D^+$ such that for each $(x_i, y_i) \in D^+$, a projection matrix maps the entity vector $\mathbf{x}_i$ to its hypernym vector $\mathbf{y}_i$. A similar model is also trained over $D^-$. Based on the two models, we estimate the prediction score and the confidence score for each $(x_i, y_i) \in D^U$. In the transductive learning stage, a joint optimization problem is formed to learn the final prediction score for each $(x_i, y_i) \in D^U$. It aims to minimize the prediction errors based on the human labeled data, the initial model prediction and linguistic rules. It also employs non-linear mappings to capture linguistic regularities of *is-a* relations other than linear projections.

### 3.2 Initial Model Training

The initial stage models how entities are translated to their hypernyms or non-hypernyms by projection learning. We first train a *Skip-gram* model (Mikolov et al., 2013) to learn word embeddings over a large text corpus. Inspired by

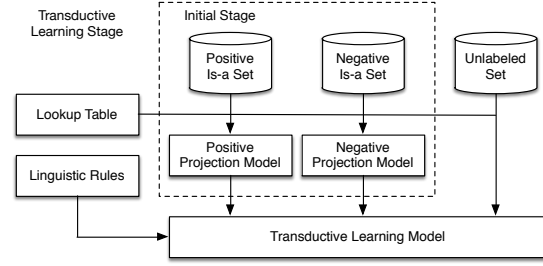

Figure 1: General framework of our approach.

(Fu et al., 2014; Wang and He, 2016), for each $(x_i, y_i) \in D^+$, we assume there is a positive projection model such that $\mathbf{M}^+\mathbf{x}_i \approx \mathbf{y}_i$ where $\mathbf{M}^+$ is an $|\mathbf{x}_i| \times |\mathbf{x}_i|$ projection matrix[3]. However, this model does not capture the semantics of *not-is-a* relations. Thus, we learn a negative projection model $\mathbf{M}^-\mathbf{x}_i \approx \mathbf{y}_i$ where $(x_i, y_i) \in D^-$. This approach is equivalent to learning two separate translation models within the same semantic space. For parameter estimation, we minimize the two following objectives:

$$J(\mathbf{M}^+) = \frac{1}{2} \sum_{(x_i,y_i) \in D^+} \|\mathbf{M}^+\mathbf{x}_i - \mathbf{y}_i\|_2^2 + \frac{\lambda}{2}\|\mathbf{M}^+\|_F^2$$

$$J(\mathbf{M}^-) = \frac{1}{2} \sum_{(x_i,y_i) \in D^-} \|\mathbf{M}^-\mathbf{x}_i - \mathbf{y}_i\|_2^2 + \frac{\lambda}{2}\|\mathbf{M}^-\|_F^2$$

where $\lambda > 0$ is a Tikhonov regularization parameter (Golub et al., 1999).

In the testing phase, for each $(x_i, y_i) \in D^U$, denote $d^+(x_i, y_i) = \|\mathbf{M}^+\mathbf{x}_i - \mathbf{y}_i\|_2$ and $d^-(x_i, y_i) = \|\mathbf{M}^-\mathbf{x}_i - \mathbf{y}_i\|_2$. The *prediction score* is defined as:

$$score(x_i, y_i) = \tanh(d^-(x_i, y_i) - d^+(x_i, y_i))$$

where $score(x_i, y_i) \in (-1, 1)$. Higher prediction score indicates there is a larger probability of an *is-a* relation between $x_i$ and $y_i$. We choose the hyperbolic tangent function rather than the sigmoid function to avoid the widespread saturation of sigmoid function (Menon et al., 1996).

The difference between $d^+(x_i, y_i)$ and $d^-(x_i, y_i)$ can be also used to indicate whether the models are confident enough to make a prediction. In this paper, we calculate the *confidence score* as:

$$conf(x_i, y_i) = \frac{|d^+(x_i, y_i) - d^-(x_i, y_i)|}{\max\{d^+(x_i, y_i), d^-(x_i, y_i)\}}$$

---

[3]We have also examined piecewise linear projection models proposed in (Fu et al., 2014; Wang and He, 2016) as the initial models for transductive learning. However, we found that this practice is less efficient and the performance does not improve significantly.

where $conf(x_i, y_i) \in (0, 1)$. Higher confidence score means that there is a larger probability that the models can predict whether there is an *is-a* relation between $x_i$ and $y_i$ correctly. This score gives different data instances different weights in the transductive learning stage.

### 3.3 Transductive Non-linear Learning

Although linear projection methods are effective for Chinese hypernym prediction, it does not encode non-linear transformation and only leverages the positive data. We present an optimization framework for non-linear mapping utilizing both labeled and unlabeled data and linguistic rules by transductive learning (Gammerman et al., 1998; Chapelle et al., 2006).

Let $F_i$ be the final prediction score of the word/phrase pair $(x_i, y_i)$. In the initialization stage of our algorithm, we set $F_i = 1$ if $(x_i, y_i) \in D^+$, $F_i = -1$ if $(x_i, y_i) \in D^-$ and set $F_i$ randomly in $(-1, 1)$ if $(x_i, y_i) \in D^U$. In matrix representation, denote $\mathbf{F}$ as the $m \times 1$ final prediction vector where $m = |D^+| + |D^-| + |D^U|$. $F_i$ is the $i$th element in $\mathbf{F}$. The three components in our transductive learning model are as follows:

#### 3.3.1 Initial Prediction

Denote $\mathbf{S}$ as an $m \times 1$ initial prediction vector. We set $S_i = 1$ if $(x_i, y_i) \in D^+$, $S_i = -1$ if $(x_i, y_i) \in D^-$ and $S_i = score(x_i, y_i)$ if $(x_i, y_i) \in D^U$. In order to encode the confidence of model prediction, we define $\mathbf{W}$ as an $m \times m$ diagonal weight matrix. The element in the $i$th row and the $j$th column of $\mathbf{W}$ is set as follows:

$$W_{i,j} = \begin{cases} conf(x_i, y_i) & i = j, (x_i, y_i) \in D^U \\ 1 & i = j, (x_i, y_i) \in D^+ \cup D^- \\ 0 & \text{Otherwise} \end{cases}$$

The objective function is defined as: $\mathscr{O}_s = \|\mathbf{W}(\mathbf{F} - \mathbf{S})\|_2^2$, which encodes the hypothesis that the final prediction should be similar to the initial prediction for unlabeled data or human labeling for training data. The weight matrix $\mathbf{W}$ gives the largest weight (i.e., 1) to all the pairs in $D^+ \cup D^-$ and a larger weight to the pair $(x_i, y_i) \in D^U$ if the initial prediction is more confident.

#### 3.3.2 Linguistic Rules

Although linguistic rules can only cover a few circumstances, they are effective to guide the learning process. For Chinese hypernym prediction,

Li et al. (2015) study the word formation of conceptual categories in Chinese Wikipedia. In our model, let $C$ be the collection of linguistic rules. $\gamma_i$ is the true positive (or negative) rate with respect to the respective positive (or negative) rule $c_i \in C$, estimated over the training set. Considering the word formation of Chinese entities and hypernyms, we design one positive rule (i.e., P1) and two negative rules (i.e., N1 and N2), shown in Table 1.

Let $\mathbf{R}$ be an $m \times 1$ linguistic rule vector and $R_i$ is the $i$th element in $\mathbf{R}$. For training data, we set $R_i = 1$ if $(x_i, y_i) \in D^+$ and $R_i = -1$ if $(x_i, y_i) \in D^-$, which follows the same settings as those in $\mathbf{S}$. For unlabeled pairs that do not match any linguistic rules in $C$, we update $R_i = F_i$ in each iteration of the learning process, meaning no loss for errors imposed in this part.

For other conditions, denote $C_{(x_i, y_i)} \subseteq C$ as the collection of rules that $(x_i, y_i)$ matches. If $C_{(x_i, y_i)}$ are positive rules, we set $R_i$ as follows:

$$R_i = \max\{F_i, \max_{c_j \in C_{(x_i, y_i)}} \gamma_j\}$$

Similarly, if $C_{(x_i, y_i)}$ are negative rules, we have:

$$R_i = -\max\{-F_i, \max_{c_j \in C_{(x_i, y_i)}} \gamma_j\}$$

which means $F_i$ receives a penalty only if $F_i < \max_{c_j \in C_{(x_i, y_i)}} \gamma_j$ for pairs that match positive rules or $F_i > -\max_{c_j \in C_{(x_i, y_i)}} \gamma_j$ for negative rules[4]. The objective function is: $\mathscr{O}_r = \|\mathbf{F} - \mathbf{R}\|_2^2$. In this way, our model can integrate arbitrary "soft" constraints, making it robust to false positives or negatives introduced by these rules.

#### 3.3.3 Non-linear Learning

*TransLP* is a transductive label propagation framework (Liu and Yang, 2015) for link prediction, previously used for applications such as text classification (Xu et al., 2016). In our work, we extend their work for our task, modeling non-linear mappings from entities to hypernyms.

For *is-a* relations, we find that if $y$ is the hypernym of $x$, it is likely that $y$ is the hypernym of entities that are semantically close to $x$. For example,

---

[4]We do not consider the cases where a pair matches both positive and negative rules because such cases are very rare, and even non-existent in our datasets. However, our method can deal with these cases by using some simple heuristics. For example, we can update $R_i$ using either of the following two ways: i) $R_i = F_i$ and ii) $R_i = F_i + \sum_{c_j \in C_{(x_i, y_i)}} \gamma_j$.

| | | |
|---|---|---|
| **P1** | The head word of the entity $x$ matches that of the candidate hypernym $y$. For example, 动物 (Animal) is the correct hypernym of 哺乳动物 (Mammal). |
| **N1** | The head word of the entity $x$ matches the non-head word of the candidate hypernym $y$. For example, 动物学 (Zoology) is not a hypernym of 哺乳动物 (Mammal). |
| **N2** | The head word of the candidate hypernym $y$ matches an entry in a Chinese lexicon extended based on the lexicon used in Li et al., (2015). It consists of 184 non-taxonomic, thematic words such as 政治(Politics), 军事(Military), etc. |

Table 1: Three linguistic rules used in our work for Chinese hypernym prediction.

if we know `United States` is a `country`, we can infer `country` is the hypernym of similar entities such as `Canada`, `Australia`, etc. This intuition can be encoded in the similarity of the two pairs $p_i = (x_i, y_i)$ and $p_j = (x_j, y_j)$:

$$\text{sim}(p_i, p_j) = \begin{cases} \cos(\mathbf{x}_i, \mathbf{x}_j) & y_i = y_j \\ 0 & \text{otherwise} \end{cases} \quad (1)$$

where $\mathbf{x}_i$ is the embedding vector of $x_i$[5].

This similarity indicates there exists a non-linear mapping from entities to hypernyms, which can not be encoded in linear projection based methods (Fu et al., 2014; Wang and He, 2016). Based on *TransLP* (Liu and Yang, 2015), this intuition can be model as propagating class labels (*is-a* or *not-is-a*) of labeled word/phrase pairs to similar unlabeled ones based on Eq. (1). For example, the score of *is-a* relations between `United State` and `country` will propagate to pairs such as (`Canada`, `country`) and (`Australia`, `country`) by random walks.

Denote $\mathbf{F}^*$ as the optimal solution of the problem $\min \mathscr{O}_s + \mathscr{O}_r$. Inspired by (Liu and Yang, 2015; Xu et al., 2016), we can add a Gaussian prior $N(\mathbf{F}^*, \mathbf{\Sigma})$ to $\mathbf{F}$ where $\mathbf{\Sigma}$ is the covariance matrix and $\Sigma_{i,j} = \text{sim}(p_i, p_j)$. Hence the optimization objective of this part is defined as: $\mathscr{O}_n = \mathbf{F}^T \mathbf{\Sigma}^{-1} \mathbf{F}$ which is linearly proportional to the negative likelihood of the Gaussian random field prior. This means we minimize the training error and encourage $\mathbf{F}$ to have a smooth propagation with respect to the similarities among pairs defined by Eq. (1) at the same time.

### 3.3.4 Joint Optimization

Based on the three components, we minimize the following function:

$$J(\mathbf{F}) = \mathscr{O}_s + \mathscr{O}_r + \frac{\mu_1}{2}\mathscr{O}_n + \frac{\mu_2}{2}\|\mathbf{F}\|_2^2 \quad (2)$$

---

[5]We only consider the similarity between entities and not candidate hypernyms because the similar rule for candidate hypernyms is not true. For example, nouns close to `country` in our *Skip-gram* model are `region`, `department`, etc. They are not all correct hypernyms of `United States`, `Canada`, `Australia`, etc.

where $\|\mathbf{F}\|_2^2$ imposes an additional smooth $l_2$-regularization on $\mathbf{F}$. $\mu_1$ and $\mu_2$ are tuning parameters.

A basic approach to learn the optimal values of $\mathbf{F}$ is via gradient descent. Based on Eq. (2), the derivative of $\mathbf{F}$ with respect to $J(\mathbf{F})$ can be computed as follows:

$$\frac{\mathrm{d}J(\mathbf{F})}{\mathrm{d}\mathbf{F}} = \mathbf{W}^2(\mathbf{F}-\mathbf{S}) + (\mathbf{F}-\mathbf{R}) + \mu_1\mathbf{\Sigma}^{-1}\mathbf{F} + \mu_2\mathbf{F}$$

Optimizing Eq. (2) is computationally expensive when $m$ is large. After $\mathbf{W}^2$, $\mathbf{S}$, $\mathbf{R}$ and $\mathbf{\Sigma}^{-1}$ are pre-computed, the runtime complexity of the loop of gradient descent is $O(tm^2)$ where $t$ is the number of iterations.

To speed up the learning process, we introduce a *blockwise gradient descent* technique. From the definition of Eq. (2), we can see that the optimal values of $F_i$ and $F_j$ with respect to $(x_i, y_i)$ and $(x_j, y_j)$ are irrelevant if $y_i \neq y_j$. Therefore, the original optimization problem can be decomposed and solved separately according to different candidate hypernyms.

Let $H$ be the collection of candidate hypernyms in $D^U$. For each $h \in H$, denote $D_h$ as the collection of word/phase pairs in $D^+ \cup D^- \cup D^U$ that share the same candidate hypernym $h$. The original problem can be decomposed into $|H|$ optimization subproblems over $D_h$ for each $h \in H$. The runtime complexity is $O(\sum_{h \in D_h} t_h|D_h|^2)$ where $t_h$ is the number of iterations to solve the subproblem over $D_h$. Although we do not know the upper bounds on the numbers of iterations of these two learning techniques, the runtime complexity can be reduced by blockwise gradient descent for two reasons: i) $\sum_{h \in D_h} |D_h| \leq m$ and ii) $t_h$ has a large probability to be smaller than $t$ due to the smaller number of data instances. This technique can be also viewed as optimizing Eq. (2) based on blockwise matrix computation.

Finally, for each $(x_i, y_i) \in D^U$, we predict that $y_i$ is a hypernym of $x_i$ if $F_i > \theta$ where $\theta \in (-1, 1)$ is a threshold tuned on the development set.

## 4 Experiments

In this section, we conduct experiments to evaluate our method. Section 4.1 to Section 4.5 report the experimental steps on Chinese datasets. We present the performance on English datasets in Section 4.6 and a discussion in Section 4.7.

### 4.1 Experimental Data

The Chinese text corpus is extracted from the contents of 1.2M entity pages from Baidu Baike[6], a Chinese online encyclopedia. It contains approximately 1.1B words. We use the open source toolkit *Ansj*[7] for Chinese word segmentation.

We have two collections of Chinese word/phase pairs as ground truth datasets. Each pair is labeled with an *is-a* or *not-is-a* tag. The first one (denoted as *FD*) is from Fu et al., (2014), containing 1,391 *is-a* pairs and 4,294 *not-is-a* pairs, which is the first publicly available dataset to evaluate this task. The second one (denoted as *BK*) is larger in size and crawled from Baidu Baike by ourselves, consisting of <entity, category> pairs. For each pair in *BK*, we ask multiple human annotators to label the tag and discard the pair with inconsistent labels by different annotators. In total, it contains 3,870 *is-a* pairs and 3,582 *not-is-a* pairs[8].

In the following experiments, we use 60% of the data for training, 20% for development and 20% for testing, partitioned randomly. By rotating the 5-fold subsets of the datasets, we report the performance of each method on average.

### 4.2 Parameter Analysis

The word embeddings are pre-trained by ourselves on the Chinese corpus. In total, we obtain the 100-dimensional embedding vectors of 5.8M distinct words. The regularization parameters are set to $\lambda = 10^{-3}$ and $\mu_1 = \mu_2 = 10^{-4}$, fine tuned on the development set.

The choice of $\theta$ reflects the precision-recall trade-off in our model. A larger value of $\theta$ means we pay more attention to precision rather than recall. Figure 2 illustrates the precision-recall curves on both datasets. It can be seen that the performance of our method is generally better in *BK* than *FD*. The most probable cause is that *BK* is a large dataset with more "balanced" numbers of positive

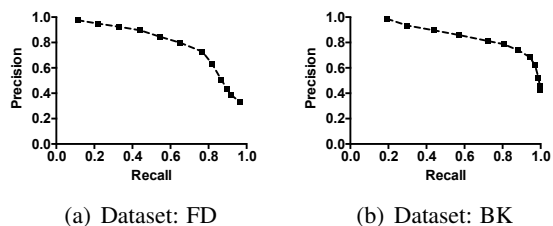

(a) Dataset: FD        (b) Dataset: BK

Figure 2: Precision-recall curve with respect to the tuning of $\theta$ on development sets (%).

and negative data. Finally, $\theta$ is set to 0.05 on *FD* and 0.1 on *BK*.

### 4.3 Performance

In a series of previous work (Fu et al., 2013, 2014; Wang and He, 2016), several pattern-based, inference-based and encyclopedia-based *is-a* relation extraction methods for English have been implemented for the Chinese language. As their experiments show, these methods achieve the F-measure of lower than 60% in most cases, which are not suggested to be strong baselines for Chinese hypernym prediction. Interested readers may refer to their papers for the experimental results.

To make the convincing conclusion, we employ two recent state-of-the-art approaches for Chinese *is-a* relation identification (Fu et al., 2014; Wang and He, 2016) as baselines. We also take the word embedding based classification approach (Mirza and Tonelli, 2016)[9] and Chinese Wikipedia based SVM model (Li et al., 2015) as baselines to predict *is-a* relations between words[10]. The experimental results are illustrated in Table 3.

For Fu et al., (2014), we test the performance using a linear projection model (denoted as S in Table 3) and piecewise projection models (P). It shows that the semantics of *is-a* relations are better modeled by multiple projection models, with a slightly improvement in F-measure. By combining iterative projection models and pattern-based validation, the most recent approach (Wang and He, 2016) increases the F-measure by 4% and 2% in two datasets. In this method, the pattern-based statistics are calculated using the same cor-

---

[6]https://baike.baidu.com/
[7]https://github.com/NLPchina/ansj_seg/
[8]Our dataset is publicly available. See submitted dataset.

[9]Although the experiments in their paper are mostly related to temporal relations, the method can be applied to *is-a* relations without modification.
[10]Previously, these methods used different knowledge sources to train models and thus the results in their papers are not directly comparable with ours. To make fair comparison, we take the training data as the same knowledge source to train models for all methods.

| Candidate Hypernym | P | T | Candidate Hypernym | P | T |
|---|---|---|---|---|---|
| **Entity**: 乙烯(Ethylene) | | | **Entity**: 孙燕姿(Stefanie Sun) | | |
| 化学品(Chemical) | √ | √ | 歌手(Singer) | √ | √ |
| 有机化学(Organic Chemistry) | × | × | 明星(Star) | √ | √ |
| 有机物(Organics) | √ | √ | 人物(Person) | √ | √ |
| 气体(Gas) | √ | √ | 金曲奖 (Golden Melody Award) | √ | × |
| 自然科学(Natural Science) | × | × | 音乐人(Musician) | √ | √ |
| **Entity**: 显卡(Graphics Card) | | | **Entity**: 核反应堆(Nuclear Reactor) | | |
| 硬件(Hardware) | √ | √ | 建筑学(Architecture) | × | × |
| 电子产品(Electronic Product) | √ | √ | 核科学(Nuclear Science) | × | × |
| 电脑硬件(Computer Hardware) | √ | √ | 核能 (Nuclear Energy) | √ | × |
| 数码(Digit) | × | × | 自然科学(Natural Science) | × | × |

Table 2: Examples of model prediction. (P: prediction result, T: ground truth, √: positive, ×: negative)

| Dataset | FD | | |
|---|---|---|---|
| Method | P | R | F |
| Fu et al., (2014) (S) | 64.1 | 56.0 | 59.8 |
| Fu et al., (2014) (P) | 66.4 | 59.3 | 62.6 |
| Li et al., (2015) | 54.3 | 38.4 | 45.0 |
| Mirza and Tonelli, (2016) (C) | 67.7 | **75.2** | 69.7 |
| Mirza and Tonelli, (2016) (A) | 65.3 | 60.7 | 62.9 |
| Mirza and Tonelli, (2016) (S) | 71.9 | 60.6 | 65.7 |
| Wang and He, (2016) | 69.3 | 64.5 | 66.9 |
| Ours (Initial) | 70.7 | 69.2 | 69.9 |
| Ours | **72.8** | 70.5 | **71.6** |
| Dataset | BK | | |
| Method | P | R | F |
| Fu et al., (2014) (S) | 71.4 | 64.8 | 67.9 |
| Fu et al., (2014) (P) | 72.7 | 67.5 | 70.0 |
| Li et al., (2015) | 61.2 | 47.5 | 53.5 |
| Mirza and Tonelli, (2016) (C) | 80.3 | 75.9 | 78.0 |
| Mirza and Tonelli, (2016) (A) | 72.7 | 65.6 | 68.9 |
| Mirza and Tonelli, (2016) (S) | 78.4 | 60.7 | 68.4 |
| Wang and He, (2016) | 73.9 | 69.8 | 71.8 |
| Ours (Initial) | 81.7 | 78.5 | 80.0 |
| Ours | **83.6** | **80.6** | **82.1** |

Table 3: Performance comparison on test sets for Chinese hypernym prediction (%).

pus over which we train word embedding models. The main reason of the improvement may be that the projection models have a better generalization power by applying an iterative learning paradigm.

Mirza and Tonelli, (2016) is implemented using three different strategies in combining the word vectors of a pair: i) concatenation $\mathbf{x}_i \oplus \mathbf{y}_i$ (denoted as C), ii) addition $\mathbf{x}_i + \mathbf{y}_i$ (A) and iii) subtraction $\mathbf{x}_i - \mathbf{y}_i$ (S). As seen, the classification models using addition and subtraction have similar performance in two datasets, while the concatenation strategy outperforms previous two approaches. Although Li et al., (2015) achieve a high performance in their dataset, this method does not perform well in ours. The most case is that the features in that work are designed specifically for the Chinese Wikipedia category system. Our initial model has a higher accuracy than all the base-

| TP/TN Rate | Rule P1 | Rule N1 | Rule N2 |
|---|---|---|---|
| Dataset FD | 98.6 | 92.3 | 94.1 |
| Dataset BK | 97.6 | 96.8 | 97.3 |

Table 4: TP/TN rates of three linguistic rules (%).

lines. By utilizing the transductive learning framework, we boost the F-measure by 1.7% and 2.1%, respectively. Therefore, our method is effective to predict hypernyms of Chinese entities.

### 4.4 Effectiveness of Linguistic Rules

To illustrate the effectiveness of linguistic rules, we present the true positive (or negative) rate by using one positive (or negative) rule solely, shown in Table 4. These values serve as $\gamma_i$s in the transductive learning stage. The results indicate that these rules have high precision (over 90%) over both datasets for our task.

We state that currently we only use a few hand-craft linguistic rules in our work. The proposed approach is a general framework that can encode arbitrary numbers of rules of any kind and in any language.

### 4.5 Error Analysis and Case Studies

We analyze correct and error cases in the experiments. Some examples of prediction results are shown in Table 2. We can see that our method is generally effective. However, some mistakes occur mostly because it is difficult to distinguish strict *is-a* and *topic-of* relations. For example, the entity Nuclear Reactor is semantically close to Nuclear Energy. The error statistics show that such kind of errors account for approximately 80.2% and 78.6% in two test sets, respectively.

Based on the literature study, we find that such problem has been also reported in (Fu et al., 2013; Wang and He, 2016). To reduce such errors, we employ the Chinese thematic lexicon based on Li

et al., (2015) in the transductive learning stage but the coverage is still limited. Two possible solutions are: i) adding more negative training data of this kind; and ii) constructing a large-scale thematic lexicon automatically from the Web.

### 4.6 Experiments on English Datasets

To examine how our method can benefit hypernym prediction for the English language, we use two standard datasets in this paper. The first one is a benchmark dataset for distributional semantic evaluation, i.e., *BLESS* (Baroni and Lenci, 2011). Because the number of pairs in *BLESS* is relatively small, we also use the *Shwartz* (Shwartz et al., 2016) dataset. In the experiments, we treat the HYPER relations as positive data (1,337 pairs) and randomly sample 30% of the RANDOM relations as negative data (3,754 pairs) in *BLESS*. To create a relatively balanced dataset, we take the random split of *Shwartz* as input and use only 30% of the negative pairs. The dataset contains 14,135 positive pairs and 16,956 negative pairs. The text corpus is English Wikipedia and the embedding vectors are set to 100 dimensions.

For comparison, we test all the baselines over English datasets except Li et al., (2015). This is because most features in Li et al., (2015) can only be used in the Chinese environment. To implement Wang and He., (2016) for English, we use the original Hearst patterns (Hearst, 1992) to perform relation selection and do not consider *not-is-a* patterns. We also take two recent DSM based approaches (Lenci and Benotto, 2012; Santus et al., 2014) as baselines. As for our own method, we do not use linguistic rules in Table 1 for English. The results are illustrated in Table 5. As seen, our method is superior to all the baselines over *BLESS*, with an F-measure of 81.9%. In *Shwartz*, while the approach (Mirza and Tonelli, 2016) has the highest F-measure of 80.1%, our method is generally comparable to theirs and outperforms others. The results suggest that although our method is not necessarily the state-of-the-art for English hypernym prediction, it has several potential applications. Refer to Section 4.7 for discussion.

### 4.7 Discussion

From the experiments, we can see that the proposed approach outperforms the state-of-the-art methods for Chinese hypernym prediction. Although the English language is not our focus, our approach still has relatively high performance.

| Dataset | BLESS | | |
|---|---|---|---|
| Method | P | R | F |
| Lenci and Benotto, (2012) | 42.8 | 38.6 | 40.6 |
| Santus et al., (2014) | 59.2 | 52.3 | 55.4 |
| Fu et al., (2014) (S) | 65.3 | 62.4 | 63.8 |
| Fu et al., (2014) (P) | 68.1 | 64.2 | 66.1 |
| Mirza and Tonelli, (2016) (C) | 79.4 | **84.1** | 81.7 |
| Mirza and Tonelli, (2016) (A) | 80.7 | 72.3 | 76.3 |
| Mirza and Tonelli, (2016) (S) | 78.0 | 81.2 | 79.6 |
| Wang and He, (2016) | 76.2 | 75.4 | 75.8 |
| Ours (Initial) | 79.3 | 76.3 | 77.7 |
| Ours | **84.4** | 79.5 | **81.9** |
| Dataset | Shwartz | | |
| Method | P | R | F |
| Lenci and Benotto, (2012) | 38.5 | 50.1 | 43.5 |
| Santus et al., (2014) | 51.2 | 71.5 | 59.6 |
| Fu et al., (2014) (S) | 65.6 | 66.1 | 65.8 |
| Fu et al., (2014) (P) | 62.3 | 71.9 | 67.3 |
| Mirza and Tonelli, (2016) (C) | 79.3 | **80.9** | 80.1 |
| Mirza and Tonelli, (2016) (A) | 79.1 | 79.6 | 79.4 |
| Mirza and Tonelli, (2016) (S) | **80.5** | 77.5 | 79.0 |
| Wang and He, (2016) | 75.1 | 76.3 | 75.6 |
| Ours (Initial) | 77.2 | 76.8 | 77.0 |
| Ours | 79.1 | 77.5 | 78.3 |

Table 5: Performance comparison on test sets for English hypernym prediction (%).

Additionally, our work has potential values for the following applications:[11]

**Domain-specific or Context-sparse Relation Extraction.** If the task is to predict relations between words when it is related to a specific domain or the contexts are sparse, even for English, traditional pattern-based methods are likely to fail. Our method can predict the existence of relations without explicit textual patterns and requires a relatively small amount of pairs as training data.

**Under-resourced Language Learning.** Our method can be adapted for relation extraction in languages with flexible expressions, few knowledge resources and/or low-performance NLP tools. This is because it does not require deep NLP parsing of sentences in a text corpus.

## 5 Conclusion

In summary, this paper introduces a transductive learning approach for Chinese hypernym prediction. By modeling linear projection models, linguistic rules and non-linear mappings, our method is able to identify Chinese hypernyms with high accuracy. Experiments show that the performance of our method outperforms previous approaches. We also discuss the potential applications of our method besides Chinese hypernym prediction.

---

[11]The implementations of these applications are beyond the scope of this paper.

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
