# Peer review of "Transductive Non-linear Learning for Chinese Hypernym Prediction"

_ACL 2017 — decision unknown_

[Official Review · Reviewer 1 · rating 4 · confidence 4]
soundness 5 · originality 3 · clarity 5 · impact 4 · substance 4 · appropriateness 5 · meaningful comparison 5 · presentation format Poster

The paper is clearly written, and the claims are well-supported.  The Related
Work in particular is very thorough, and clearly establishes where the proposed
work fits in the field.

I had two main questions about the method: (1) phrases are mentioned in section
3.1, but only word representations are discussed.  How are phrase
representations derived?
(2) There is no explicit connection between M^+ and M^- in the model, but they
are indirectly connected through the tanh scoring function.  How do the learned
matrices compare to one another (e.g., is M^- like -1*M^+?)?  Furthermore, what
would be the benefits/drawbacks of linking the two together directly, by
enforcing some measure of dissimilarity?

Additionally, statistical significance of the observed improvements would be
valuable.

Typographical comments:
- Line 220: "word/phase pair" should be "word/phrase pair"
- Line 245: I propose an alternate wording: instead of "entities are translated
to," say "entities are mapped to".  At first, I read that as a translation
operation in the vector space, which I think isn't exactly what's being
described.
- Line 587: "slightly improvement in F-measure" should be "slight improvement
in F-measure"
- Line 636: extraneous commas in citation
- Line 646: "The most case" should be "The most likely case" (I'm guessing)
- Line 727: extraneous period and comma in citation

[Official Review · Reviewer 2 · rating 4 · confidence 3]
soundness 5 · originality 3 · clarity 4 · impact 4 · substance 4 · appropriateness 5 · meaningful comparison 4 · presentation format Poster

- Strengths:

1. Interesting research problem
2. The method in this paper looks quite formal.
3. The authors have released their dataset with the submission.
4. The design of experiments is good.

- Weaknesses:

1. The advantage and disadvantage of the transductive learning has not yet
discussed.

- General Discussion:

In this paper, the authors introduce a transductive learning approach for
Chinese hypernym prediction, which is quite interesting problem. The authors
establish mappings from entities to hypernyms in the embedding space directly,
which sounds also quite novel. This paper is well written and easy to follow.
The first part of their method, preprocessing using embeddings, is widely used
method for the initial stage. But it's still a normal way to preprocess the
input data. The transductive model is an optimization framework for non-linear
mapping utilizing both labeled and unlabeled data. The attached supplementary
notes about the method makes it more clear. The experimental results have shown
the effectiveness of the proposed method in this paper. The authors also
released dataset, which contributes to similar research for other researchers
in future.